

# Zero-bias conductance peaks at zero applied magnetic field due to stray fields from integrated micromagnets in hybrid nanowire quantum dots

Y. Jiang[1], M. Gupta[2], C. Riggert[2], M. Pendharkar[3], C. Dempsey[3], J. S. Lee[4], S. D. Harrington[5], C. J. Palmstrøm[3,4,5], V. S. Pribiag[2] and S. M. Frolov[1*]

**1** Department of Physics and Astronomy, University of Pittsburgh,
Pittsburgh, Pennsylvania 15260, USA
**2** School of Physics and Astronomy, University of Minnesota Twin Cities,
Minneapolis, Minnesota 55455, USA
**3** Electrical and Computer Engineering, University of California,
Santa Barbara, CA, 93106, USA
**4** California NanoSystems Institute, University of California Santa Barbara,
Santa Barbara, CA, 93106, USA
**5** Materials Department, University of California Santa Barbara,
Santa Barbara, CA, 93106, USA

* frolovsm@pitt.edu

## Abstract

Many recipes for realizing topological superconductivity rely on broken time-reversal symmetry, which is often attained by applying a substantial external magnetic field. Alternatively, using magnetic materials can offer advantages through low-field operation and design flexibility on the nanoscale. Mechanisms for lifting spin degeneracy include exchange coupling, spin-dependent scattering, spin injection – all requiring direct contact between the bulk or induced superconductor and a magnetic material. Here, we implement locally broken time-reversal symmetry through dipolar coupling from nearby micromagnets to superconductor-semiconductor hybrid nanowire devices. Josephson supercurrent is hysteretic due to micromangets switching. At or around zero external magnetic field, we observe an extended presence of Andreev bound states near zero voltage bias. We also show a zero-bias peak plateau of a non-quantized value. Our findings largely reproduce earlier results where similar effects were presented in the context of topological superconductivity in a homogeneous wire, and attributed to more exotic time-reversal breaking mechanisms [1]. In contrast, our stray field profiles are not designed to create Majorana modes, and our data are compatible with a straightforward interpretation in terms of trivial states in quantum dots. At the same time, the use of micromagnets in hybrid superconductor-semiconductor devices shows promise for future experiments on topological superconductivity.

## 1  Introduction

Hybrid superconductor-semiconductor devices are developed for the experimental search of induced topological superconductivity and Majorana bound states (MBSs) [2–6]. Inspired by theory [7, 8], typical devices feature a semiconductor nanowire with strong spin-orbit interaction and a superconducting contact. Advantages this may offer for putative topological quantum computing are related to the high quality, flexibility and scalability of the hybrid platform based on well-studied and manufacturable ingredients [9, 10].

A required component for isolating individual Majorana states is a type of Zeeman interaction to lift spin degeneracy. In experiments, this is commonly an external magnetic field. Alternatively, magnetic materials can mediate spin splitting via exchange interaction from ferromagnetic insulators [1, 11] or magnetic texture from micromagnets [12–17]. They can also provide further confidence that a topological state is realized, by helping detect the inherent spin texture of its spin-helical precursor in the normal state [18–20]. These approaches come with costs such as increased device complexity, but they also offer unique opportunities for the device architecture and scalability toward circuits. For example, inhomogeneous local magnetic field profiles can be generated by proper design of micromagnets, which induce MBSs and provide methods for braiding [21].

Aside from their more exotic predicted properties such as non-Abelian exchange, MBSs are zero-energy bound states which can result in zero-bias conductance peaks (ZBP) in tunneling devices. ZBPs due to MBS are predicated to be exactly quantized under the conditions of perfect Andreev reflection, zero temperature, infinite nanowire length and total absence of disorder [22, 23]. The quantization is rapidly lost with departure from ideal conditions (see Figure 5 in ref. [22]). ZBPs were reported extensively as signatures of MBSs in experiments [2–4, 24].

Andreev bound states (ABSs) can appear in quantum dots coupled to superconductors. These are many-body states induced by the interaction of the condensate with quantum dot states. They are either localized or quasi-localized wavefunctions that do not require topological superconductivity. However, they share many experimentally observed features with MBSs, most importantly they also result in ZBPs [25]. When ABSs cross at zero bias, a ZBP may appear with a peak differential conductance that is equal to, smaller or larger than the quantized

$2e^2/h$ value. These states are referred to as non-topological or trivial. Most ZBP observations can be accounted for by considering ABS in quantum dots [26, 27]. For instance, fine-tuning of ABS-induced peak heights can be behind apparent ZBP quantization, as evidenced by the studies in non-topological regimes [6, 28].

ZBPs at zero external magnetic field have been previously reported in InAs nanowire-superconductor-magnetic insulator devices [1, 11]. Measurements were presented as evidence of MBSs at zero external magnetic field. A significant variety of explanations were put forward for how time reversal symmetry lifting was achieved and how this would lead to topological superconductivity [29]. Some theoretical work suggested that magnetic proximity cannot alone induce a topological phase transition [30–32]. However, for overlapping layers of superconductor and magnetic insulators, a topological phase may be reached with the help of electrostatic tuning [33–35]. Additionally, it has been suggested that a thin layer of magnetic insulator may work as a spin-filtering tunnel barrier [30]. These works do not incorporate stray magnetic fields into their models and do not include localized quantum dot states. In experiments [1, 11] the authors argue that the shells represent single domain micromagnets and stray fields are only present at nanowire ends. However, they do not consider the damage to magnetic shells from nanofabrication of contacts and etching of superconductor shells which can result in magnetic multidomain regimes. We shall publish a separate comment exploring this possibility.

In this paper, we integrate micromagnets with hybrid superconductor-semiconductor nanostructures based on InAs and InSb nanowires. The magnets are not in contact with the nanowire and serve as sources of dipolar magnetic and electrostatic gating fields. Hysteretic Josephson supercurrents are observed in superconductor-nanowire-superconductor (SNS) Sn-InSb-Sn junctions due to magnetization switching. In superconductor-nanowire (SN) junctions, Andreev bound states under local magnetic fields show several interesting features which reproduce behavior attributed to MBSs in other [1, 36]. In particular, zero bias peaks are observed at zero external field, the features also exhibit hysteresis in magnetic field due to magnetic switching. These zero-bias peaks exhibit some degree of stability under electric fields from magnetic and non-magnetic gate electrodes. We argue these ZBPs originate from ABS in accidental quantum dots moving closer to zero energy due to stray field suppression of the superconducting gap as well as due to direct Zeeman splitting of ABS. In some of the data, the ZBPs show plateau-like features with values close to half of conductance quantum ($G_0 = 2e^2/h$ expected for MBS in toy models), however careful exploration of the parameter space shows significant deviations from quantization.

## 2 Brief methods

We study two types of devices. The first is based on InSb nanowires with superconducting Sn shells [37]. These devices feature nanowire shadow-defined SNS Josephson junctions. For the second type of devices, we use InAs nanowires with an in-situ grown superconducting Al layer [38]. We fabricate SN junctions by wet chemical etching to remove a section of Al layer on the right side of the nanowire. This allows us to perform tunneling spectroscopy of bound states near the edge of Al. Nanowires are placed on highly doped Si wafers covered with dielectric layers of $SiO_2$ and $HfO_x$. Si is used as a global back gate to apply back-gate voltage ($V_g$) and tune the chemical potential of the nanowire, local side gates are also fabricated in some devices. Ti/Au leads are deposited to apply bias voltage or current. Our experiments are performed in dilution refrigerators, with the base temperature of approximately 50 mK.

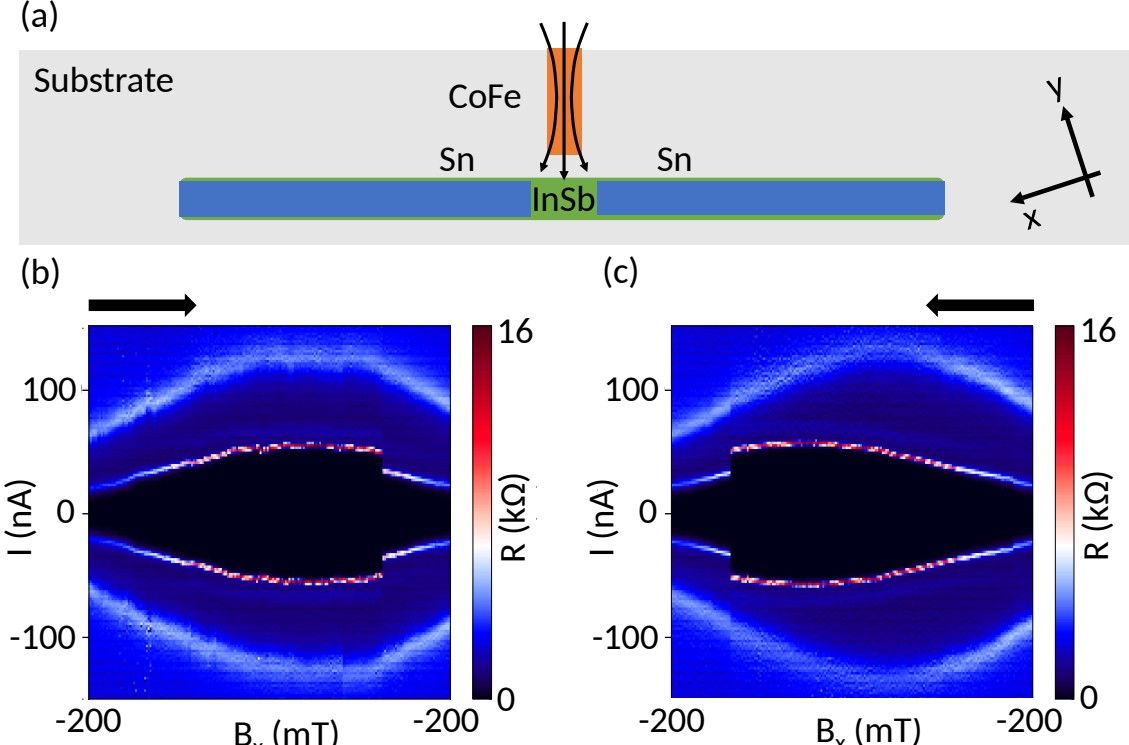

Figure 1: (a) A diagram of an InSb nanowire SNS junction. (b, c) Scans of bias current (I) vs. magnetic field ($B_x$) measured on device 1. Arrows indicate sweep directions of magnetic field. The back-gate voltage is 3 V. A known series resistance of the measurement setup is subtracted.

## 3 Results

To demonstrate the effect of local magnetic fields, we first study hysteretic supercurrents in SNS junctions. The schematic of an InSb nanowire SNS junction device is shown in Fig. 1(a). The nanowire has two superconducting Sn contacts and a window of bare InSb between the two Sn contacts. A ferromagnetic CoFe strip is deposited close to the junction. Figs. 1(b) and 1(c) show hysteretic supercurrents for two magnetic field sweep directions. The angle of applied external magnetic field is 18° with respect to the nanowire (72° with respect to CoFe strip), the switching of magnetization is observable as sharp supercurrent switches. The switching fields in these data are 124 mT ±4 mT and −136 mT ±4 mT. Device 1 happens to have two shadow junctions along the nanowire and two CoFe strips are deposited close to the two junctions. However, the data are consistent with measurements on a single junction, and we analyze it as such. A scanning electron microscope (SEM) image of device 1 and more data are available in supplementary materials. Hysteretic supercurrents in Al were reported from switching in magnetic insulator shells on InAs nanowires [1]. Those authors argued against stray magnetic fields as a mechanism. Here we show that stray fields can produce similar supercurrent switching behavior in hybrid nanowire devices.

Next, to perform tunneling spectroscopy we fabricate SN junctions using Al-InAs nanowires. These materials are closely related to those used in [1] but they do not include EuS magnetic insulator shells. Device 2 (Fig. 2(a)) is a representative SN junction device. It features two CoFe strips across the nanowire, in a configuration where they boost each other's dipolar magnetic fields. The strips can also be used as side gates ($V_{sg1}$ and $V_{sg2}$). The CoFe strips are pre-magnetized using the following procedure. First, a −0.2 T external magnetic

field is applied in the direction $\theta = 0^o$ along the long axis of CoFe strips. Then the field is swept back to zero. In some of the data, we observe that the apparent superconducting gap feature, a region of suppressed conductance around zero bias, is diminished with respect to the bulk Al value of over 200 µV, while in other data the feature is close to the presumed Al value. It is possible that magnetization in CoFe directly suppresses superconductivity in Al given the proximity to the Al shell and significant stray fields exceeding 100 mT. This, however, depends on the exact orientation of the Al shell, because the in-plane critical field of thin Al can be greater than 1T [39]. A suppressed bulk gap, like in Ref. [1] would push subgap states closer to zero ensuring prolonged ZBP regions due to level repulsion from above-gap states [25].

At zero applied field, by tuning the back gate and both side gate voltages, we observe ZBPs as shown in Fig. 2(b). This ZBP is present over a somewhat extended range in gate voltage, and therefore shares some features with ZBPs theoretically predicted for MBSs. On the other hand, at the two ends of the ZBP region, we observe peaks splitting. These features indicate that this state is more likely a trivial ABS. Similar ZBPs were reported in quantum dots in the regime between strongly coupled and weakly coupled superconducting contact where peaks can coalesce at zero bias over a finite gate voltage range [25]. We remark that the conductance at ZBP in this specific regime is larger than $G_0$, which is incompatible with MBSs in the simplest theory [23, 40].

It would be interesting to obtain MBSs at zero applied external field, however stray magnetic fields from CoFe are purposefully not designed in this experiment to realize MBSs. The fields are strongly concentrated in a narrow segment mostly overlapping with the etched region outside of the hybrid superconductor-semiconductor segment. The field is also mostly perpendicular to the nanowire and along the direction of the effective spin-orbit field. In previous work some of us suggested a micromagnetic design that could be used to create extended fields along the nanowire involving pairs of oppositely magnetized micromagnets [21].

We perform further studies of ZBPs at and around zero applied field. Figs. 2(c) and 2(d) show back and forth magnetic field sweeps with the field direction perpendicular to the wire. The data are hysteretic. The range of ZBP in field is +/- 50 mT accompanied by unidirectional abrupt switches, analogous to those seen in Fig. 1. Fig. 2(e) shows the behavior of the ZBP when the external magnetic field is applied along the wire (the CoFe strips are still premagnetized in the direction perpendicular to the nanowire). A broadened ZBP is observed. Fig. 2(f) shows the ZBP under a rotating 40 mT field (smaller than the coercive field of CoFe strips). Before this scan, the CoFe strips are also pre-magnetized the same way as for Fig. 2(b). The ZBP extends to all the rotation angles. Since there is no symmetry around zero field in Figs. 2(b-d), the data in Fig. 2(f) also do not exhibit 180° periodicity. There are no sharp state changes due to magnetic field in both Fig. 2(e) and Fig. 2(f), indicating that the CoFe strips keep their magnetization in these scans.

The conductance values of ZBPs due to ABS can have a variety of values: smaller than, larger than or equal to $G_0$. Additionally, these ZBPs may sometimes show a plateau-like feature in gate voltage or in magnetic field. In Fig. 3, we show a ZBP at zero external magnetic field with a plateau in device 2. This plateau is in gate voltage; and the external magnetic field is zero (CoFe strips are pre-magnetized). Conductance at this plateau is close to $0.5G_0$, a value consistent with chiral Majorana fermions [41, 42]. However, there is no evidence in our data that this half-quantized ZBP plateau is related to MBSs of any kind. Taken together, ZBPs in Figs. 2 and 3 show values both below and above $G_0$, while exhibiting similar behavior in bias-gate space. With enough fine-tuning, it should also be possible to obtain the same-looking features with ZBP values arbitrarily close to $G_0$ [43], for example within 5% like recently reported in Al-InAs nanowires [36].

We perform detailed micro-magnetic simulations using the numerical package Mumax3 [44, 45] to evaluate the total magnetic field, including contributions from the stray fields gen-

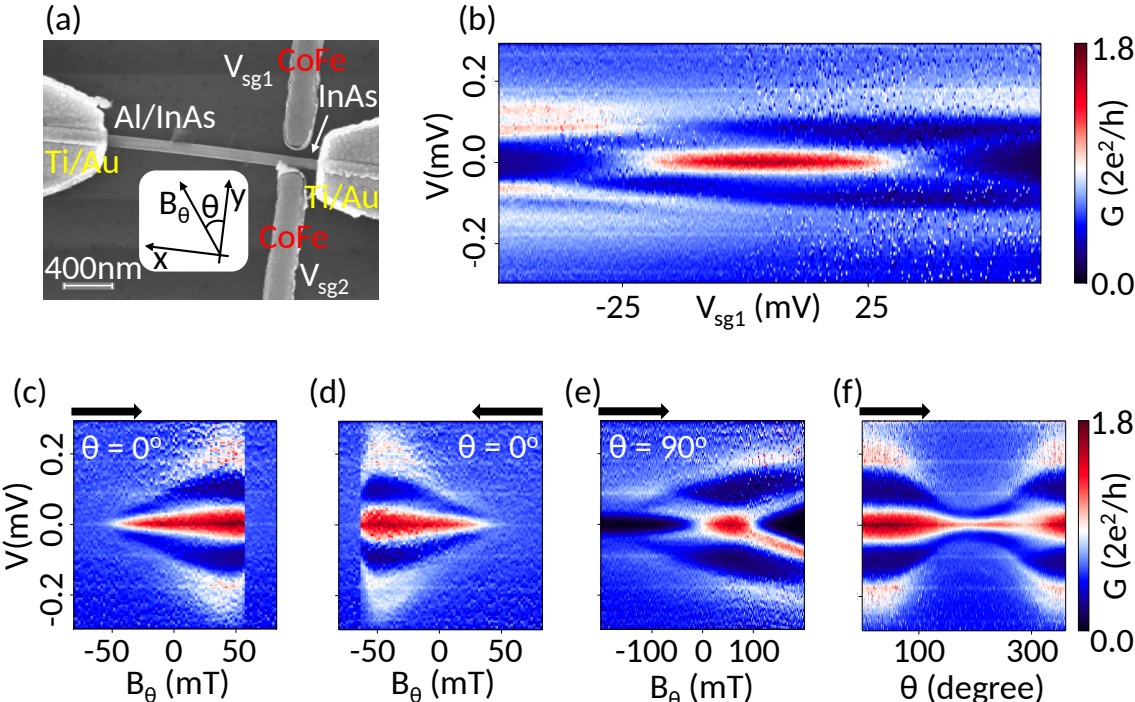

Figure 2: (a) SEM image of device 2 based on an Al-InAs nanowire. (b) Conductance as function of bias voltage (V) vs. side-gate voltage ($V_{sg1}$) at zero applied magnetic field. (c, d, e) Scans of V vs. magnetic field ($B_\theta$). $V_{sg1} = 0$ mV. The angle of the external magnetic field is labeled on the plots. (f) Scan of V vs. rotating magnetic field. The magnitude of the external magnetic field is 40 mT, $V_{sg1} = 0$. In figure 2, the back-gate voltage is at 8.25 V and the second side gate voltage ($V_{sg2}$) is at 0, the CoFe strips are magnetized along $\theta = 180°$.

erated by the magnets and the applied external magnetic field ($B_\theta$), at the tunnel barrier. We show that even at zero applied magnetic field, stray fields alone from the CoFe bar magnets are sufficient to induce ZBPs. Further, the hysteretic nature of the magnetization can explain the hysteretic nature of the observed ZBPs and critical current.

We simulate the experimental geometry shown in Fig. 2 using a cubic mesh with a size of 5 nm. The magnets are initially magnetized in the $+y$ direction by sweeping an external field from 0 to 0.12 T in the $y$-direction; the external magnetic field is then swept backwards and forwards to obtain the $y$-component of the effective field, $B_y^{eff}$, which the sum of the external and stray magnetic fields. In Fig. 4(a) we plot $B_y^{eff}$ in the plane of magnets and wire for $B_\theta = 0$ (with $\theta = 0°$) obtained during backward sweep and find that $B_y^{eff} \sim 0$ everywhere in space except in between the magnets. In the experiment, the tunnel barrier is in between the magnets, where we show stray magnetic field of magnitude $\sim 0.25$ T. Since at fields of this magnitude ZBPs have been reported in quantum dots [25], this shows that in the presence of micromagnets trivial ZBP can arise due to stray fields.

To illustrate the hysteretic behavior of these stray fields, and thus of the ZBP they induce, we average $B_y^{eff}$ over the hexagonal cross-section of the wire. We plot this averaged field, $B_y^{avg}$, for $x = 0.4$ $\mu m$ in Fig. 4(b), as a function of $B_\theta$. This shows a clear hysteresis in the effective field at the tunnel barrier. As the magnitude of the applied magnetic field increases above the coercive field of the magnets, their magnetization switches abruptly, resulting in a sudden change in $B_y^{eff}$ at the tunnel barrier in between the magnets, as shown in Figs. 4(c) and 4(d). This would result in the experimentally observed disappearance of the ZBP (Figs.

(a)

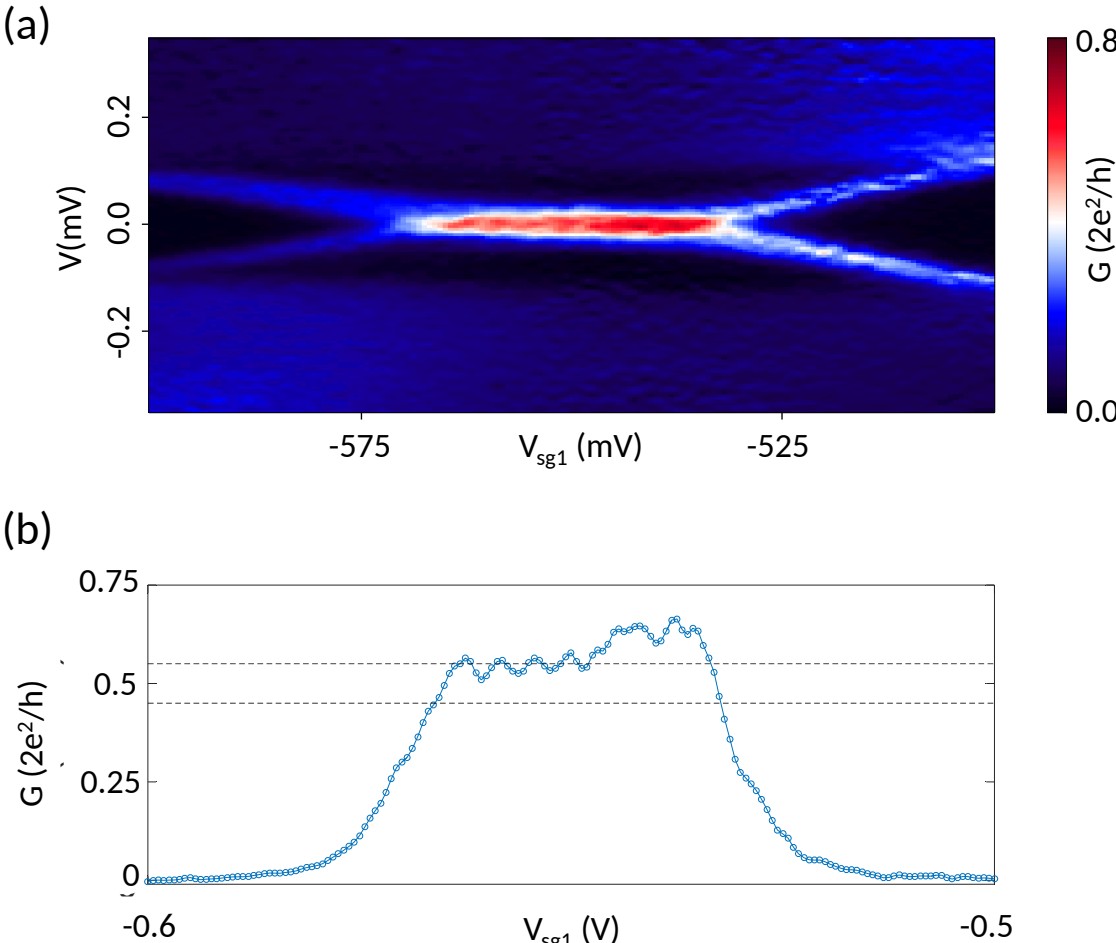

(b)

Figure 3: (a) A scan of V vs. side gate voltage ($V_{sg1}$) measured on device 2. The back-gate voltage is 6.075 V and the second side gate voltage ($V_{sg2}$) is 0 V. The CoFe strips are magnetized along $\theta = 180°$ and the external field is set to zero. (b) Line-cut at zero voltage bias from (a). Dashed lines indicate the +/- 5% range of conductance. A low-pass filter is applied for appearance.

2(c) and 2(d)) as the applied magnetic field is swept.

We have also simulated behavior of the magnetization and the effective field for field sweeps with $\theta = 90°$ and rotation of the magnetic field at a fixed magnitude. These results are shown in supplementary materials (Fig. 14). These simulations do not show any hysteretic features in the magnetic field at the tunnel barrier and are consistent with the experimental results. Additionally, we find that throughout these sweeps, a near constant magnetic field in $y$-direction is present at the junction, resulting in stable ZBPs. During the sweeps, small changes in the x-direction of the magnetic field are observed, but are too small to cause changes in the bias spectroscopy results.

In InAs nanowire systems integrating superconductors and ferromagnets, a hysteretic critical current has been used as a measure for an effective magnetic field in the absence of any applied external field [1, 11]. However, as we have shown, in the presence of ferromagnets, hysteretic critical currents can arise due to the hysteretic nature of magnetization, giving rise to a hysteretic effective field. To show this we have also simulated the double SNS junction geometry measured shown in Figs. 1(b), 1(c). We observe similar hysteresis and sharp changes

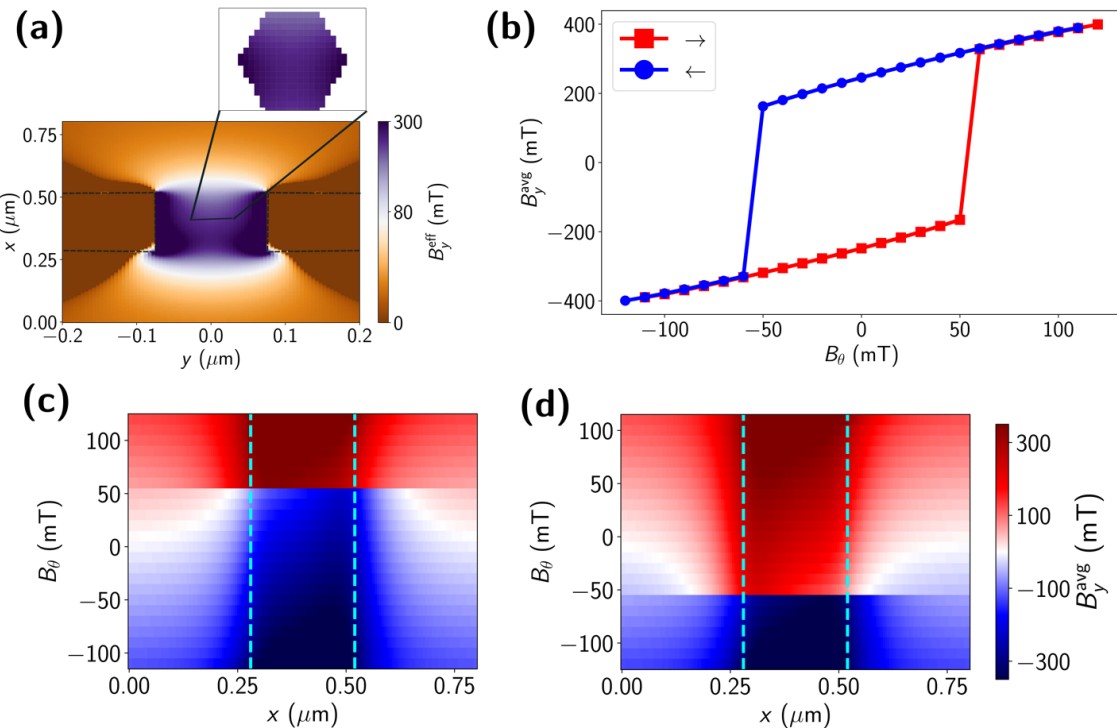

Figure 4: Micromagnetic simulations. Here $\theta = 0°$. (a) Colormap of $\mathbf{B_y^{eff}}$ as a function of spatial dimension x and y for $\mathbf{B_\theta = 0}$. Dashed lines indicate micromagnet contours. (*inset*) $\mathbf{B_y^{eff}}$ across the nanowire cross-section. (b) $\mathbf{B_y^{avg}}$ as a function of $\mathbf{B_\theta}$ for the nanowire cross-section at $\mathbf{x = 0.4}$ $\mu$m. Colormap of averaged $\mathbf{B_y^{eff}}$ across the nanowire cross-section as a function of $\mathbf{B_\theta}$ and $\mathbf{x}$ for (c) forward and (d) backward sweep. Dashed cyan lines indicate the location of the CoFe bar magnets.

in the effective magnetic field perpendicular to the current flow direction at the junction area (Figs. 14(e), 14(f)). These simulations can help understand the experimentally observed hysteretic critical current shown in Fig. 1(b), 1(c).

Experiments performed in our devices demonstrate the effectiveness of local magnetic fields generated by micromagnets in inducing strong local magnetic fields that are sufficient, in principle, to enter the topological regime, or to significantly affect the spectrum of trivial ABSs. Quantum dots commonly appear in nanowire devices, and this explanation fully accounts for ZBPs presented here and in other works. Nevertheless, our experiments and simulations demonstrate an approach to studying topological superconductivity in appropriately designed devices, and with alternative explanations in mind.

## 3.1 Methods: Micromagnetic simulations

Micromagnetic simulations were performed using Mumax3 [44]. Simulation box is discretized using a cubic mesh of size 5 nm for the tunnel barrier geometries and a 10 nm size for the double SNS junction geometry. The parameters defining the material in the simulations are saturation magnetization $M_s = 1.44 \times 10^6$ A/m, exchange stiffness $A_{ex} = 6.7 \times 10^{11}$ J/m and the damping constant $\alpha = 0.01$. The average of effective magnetic field is performed over the hexagonal cross-section of the nanowire with a diagonal width of 100 nm. This gives a one-dimensional magnetic field profile along the length of the nanowire for each value of applied field.

## 3.2 Data availability and processing steps

Input resistance or conductance is subtracted from plotted data. For device 1, the input resistance is 4420 Ω. For device 2, the input resistance is 4487 Ω. For some plots, the data are cropped on the sides to demonstrate an interesting data range. Some data are shifted in the bias axis to deal with shift from measurement setup. Readers may check all corresponding original data available on Zenodo. Mumax3 scripts for evaluating the effective field and Python code used for analysis are also available on Zenodo.

## 3.3 Duration and volume of study

This article is written based on more than 5500 datasets from 5 separate dilution refrigerator cooldowns. For each cooldown, we measure several devices. We measured 19 SN devices, and 20 SNS devices (7 of 20 devices have CoFe strips). Among them, 4 SN devices exhibit clear ABSs; 2 SNS devices show ABSs; and 4 SNS devices have measurable supercurrent. The yield is limited by fabrication issues, device quality and occasional static discharge damage to devices. Within each device, the behaviors in the main text require fine-tuning. The more general behaviors are presented in the supplementary materials.

# 4 Further reading

Introduction and reviews of Majorana physics in nanowires can be found in the references [7,8, 46–49]. MBSs induced by magnetic textures are discussed in these references [12–17,50,51]. Trivial ZBPs are discussed in these references [6,27,52].

# Acknowledgments

InSb nanowires were provided by G. Badawy, S. Gazibegovic and E. Bakkers. InAs-Al nanowires were provided by S. Khan and P. Krogstrup. We thank P. Crowell and V. Mourik for useful discussions.

**Funding information** This work is supported by the Department of Energy under award no. DE-SC-0019274. The authors acknowledge the Minnesota Supercomputing Institute (MSI) at the University of Minnesota for providing resources that contributed to the research results reported within this paper. URL: http://www.msi.umn.edu.

**Author contributions** YJ performed device fabrication and measurements. YJ and SMF analyzed the experimental data. MG, CR and VSP performed and analyzed the micromagnetic simulations. MP, CD, JSL, SH and CP grew Sn shells. YJ, MG, CR, SF and VSP wrote the manuscript, with input from all authors.

# A   Supplementary materials

(a)

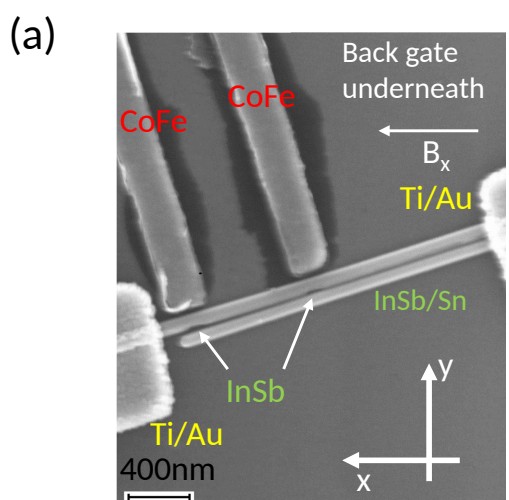

(b)   Double S-N-S junctions        (c)        Double S-N-S junctions

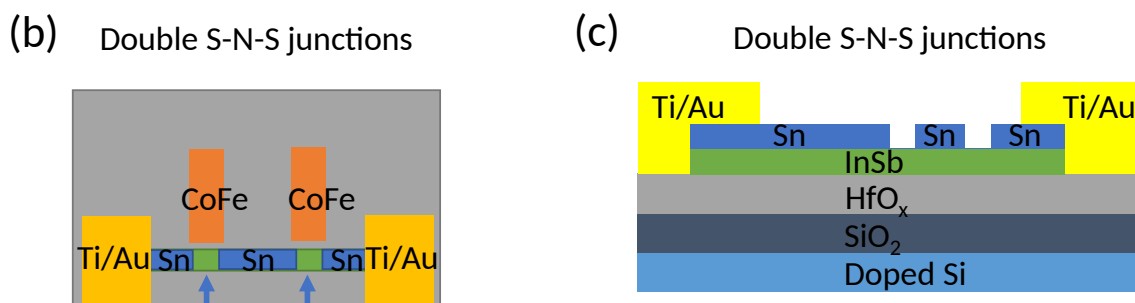

Figure 5: An InSb nanowire device, device 1, with two SNS junctions. (a) An SEM image of device 1. The direction of the applied magnetic field is given by the arrow. (b) Top-view schematic of device 1. (c) Cross-section of the double SNS junctions.

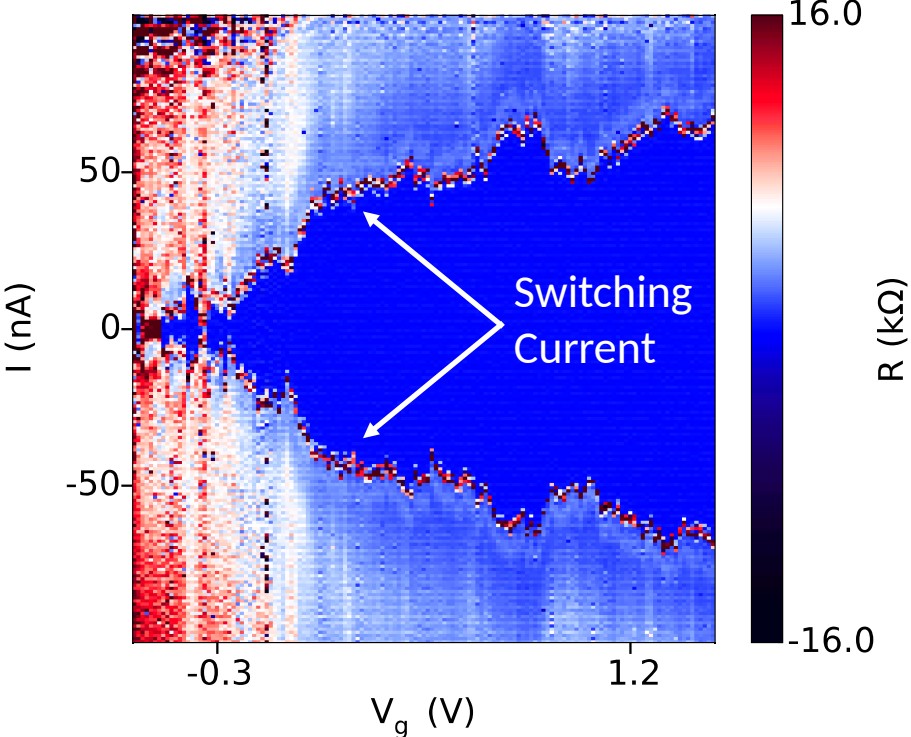

Figure 6: 2D plot of bias current (I) vs gate voltage ($V_g$) in device 1. Resistance is acquired by taking differential of measured voltage and subtracting input resistance. The switching current is indicated by arrows.

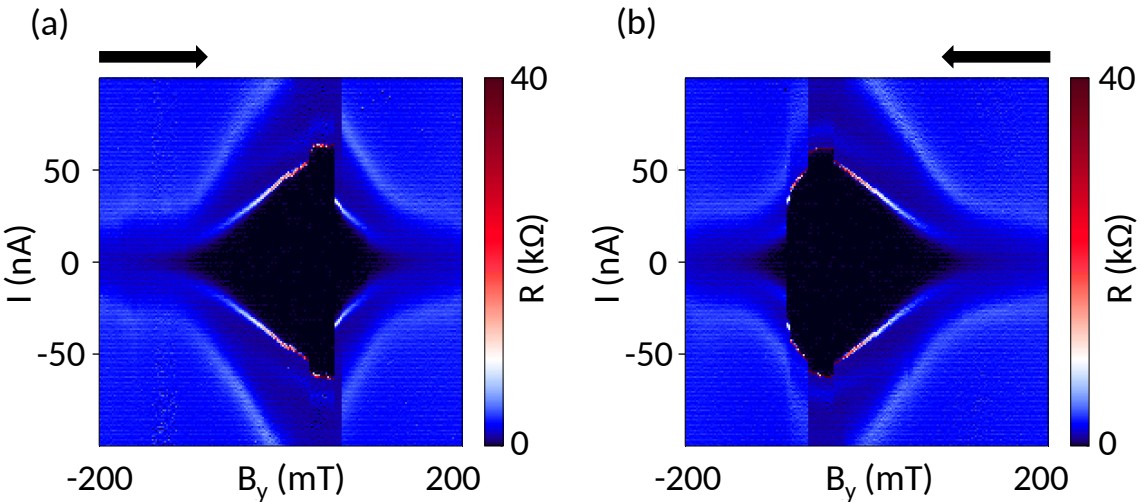

Figure 7: 2D plots of bias current (*I*) with respect to magnetic field ($B_y$) in device 1. $V_g = 3$ V. (a) $B_y$ is swept from -0.2 T to 0.2 T. (b) $B_y$ is swept from 0.2 T to -0.2 T.

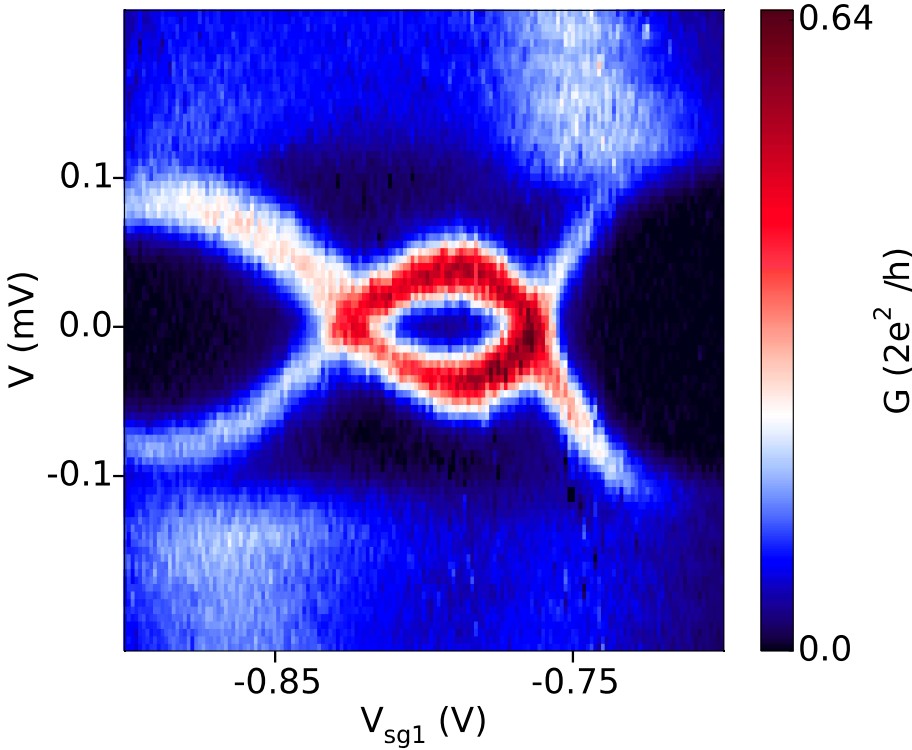

Figure 8: Scans of $V$ vs. side gate voltage ($V_{sg1}$) measured on device 2. $V_g = 6.45$ V. $V_{sg2} = 0$ V. Data are taken at zero external magnetic field. The CoFe strips are magnetized using the pre-magnetization process described in Figure 2(b).

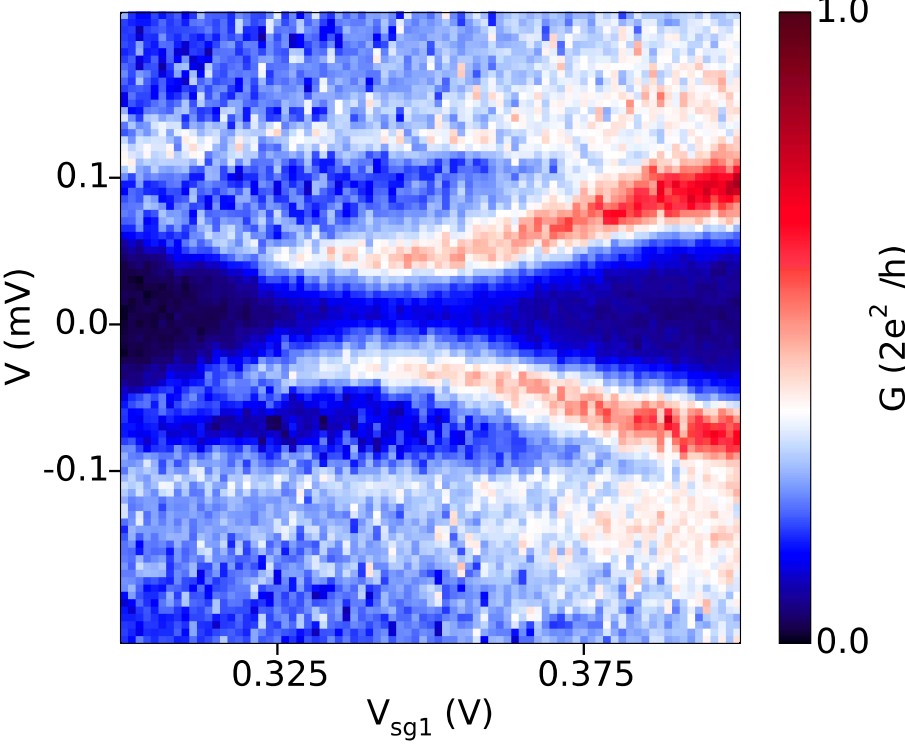

Figure 9: Scans of $V$ vs. side gate voltage ($V_{sg1}$) measured on device 2. $V_g = 6.075$ V. $V_{sg2} = 0$ V. Data are taken at zero external magnetic field. The CoFe strips are magnetized using the pre-magnetization process described in Figure 2(b).

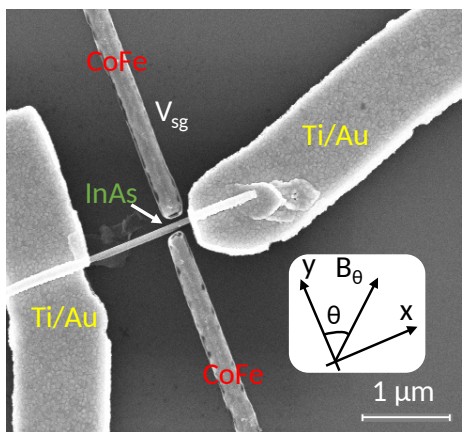

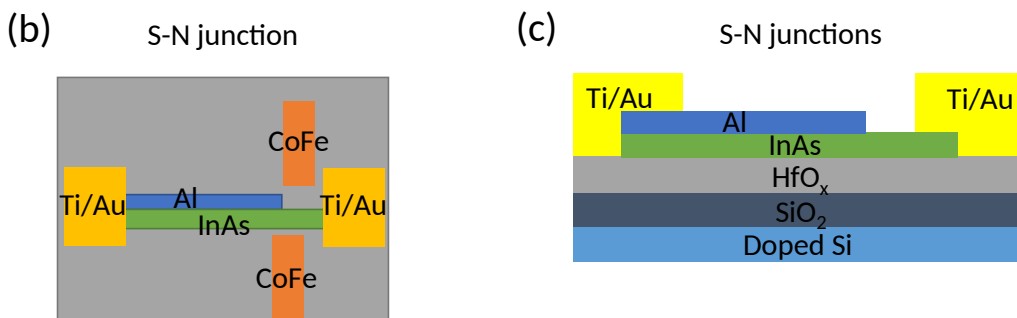

Figure 10: Another InAs nanowire device, device 3, with one SN junction and two CoFe strips. (a) An SEM image of device 3. The upper CoFe strip is also used as a side gate ($V_{sg}$). (b) Top-view schematic of device 3. (c) Cross-section schematic of the SN junction.

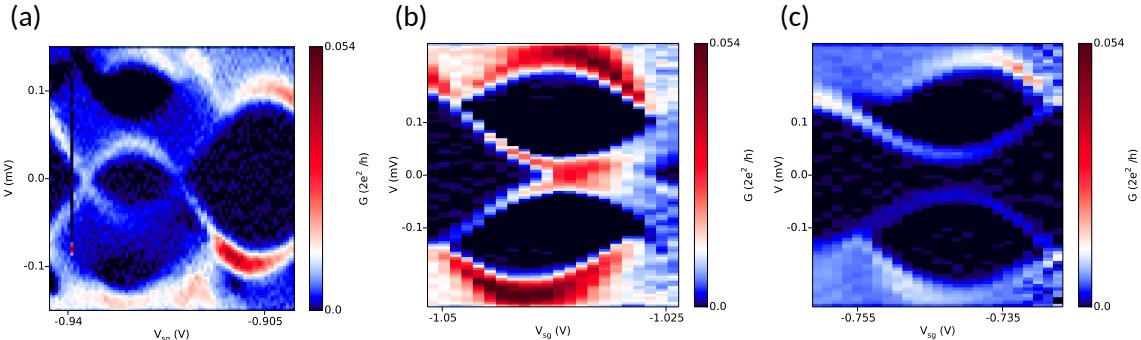

Figure 11: 2D plots of voltage bias ($V$) with respect of side-gate voltage ($V_{sg}$) in device 3. Conductance is acquired by taking a derivative of measured current. The CoFe strips are pre-magnetized by applying a -0.2 T magnetic field $B_\theta$ where $\theta = 46^\circ$. After pre-magnetization, the external magnetic field is swept back to zero for taking the data. (a) $V_g = 9$ V. Weak-coupled case. (b) $V_g = 9.75$ V. Between weak-coupled and strong-coupled. (c) $V_g = 9$ V. Strong-coupled case.

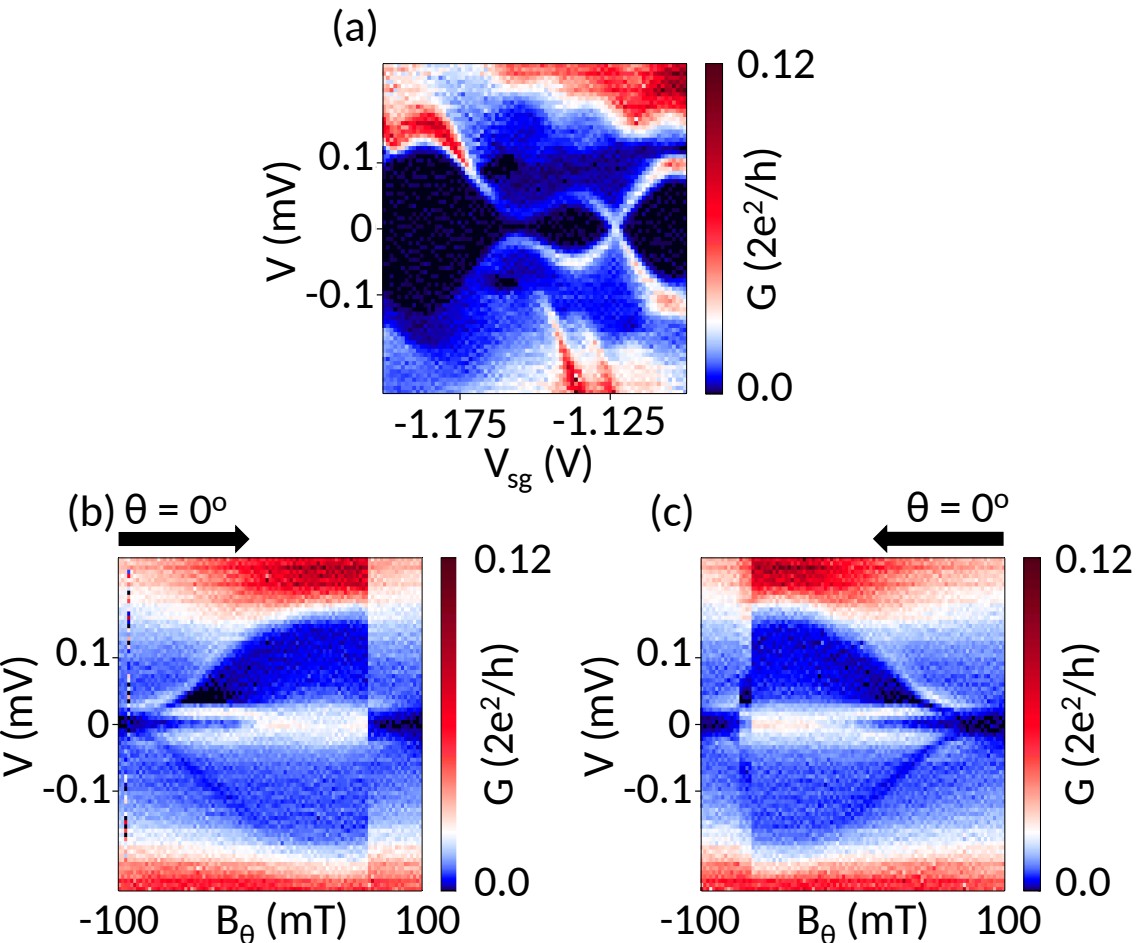

Figure 12: ZBP and hysteresis under magnetic field scans. $V_g = 10.5$ V measured in device 3. (a) 2D plots of voltage bias ($V$) with respect of side-gate voltage ($V_{sg}$) in device 3. Conductance is acquired by taking differential of measured current. The CoFe strips are pre-magnetized by applying a -0.1 T magnetic field $B_\theta$ where $\theta = 0^o$. After pre-magnetization, the external magnetic field is swept back to zero for taking the data. (b, c) 2D plots of voltage bias ($V$) with respect to the external magnetic field, $B_\theta$ ($\theta = 0^o$) in device 3. Sweeping direction of magnetic field is given by arrows.

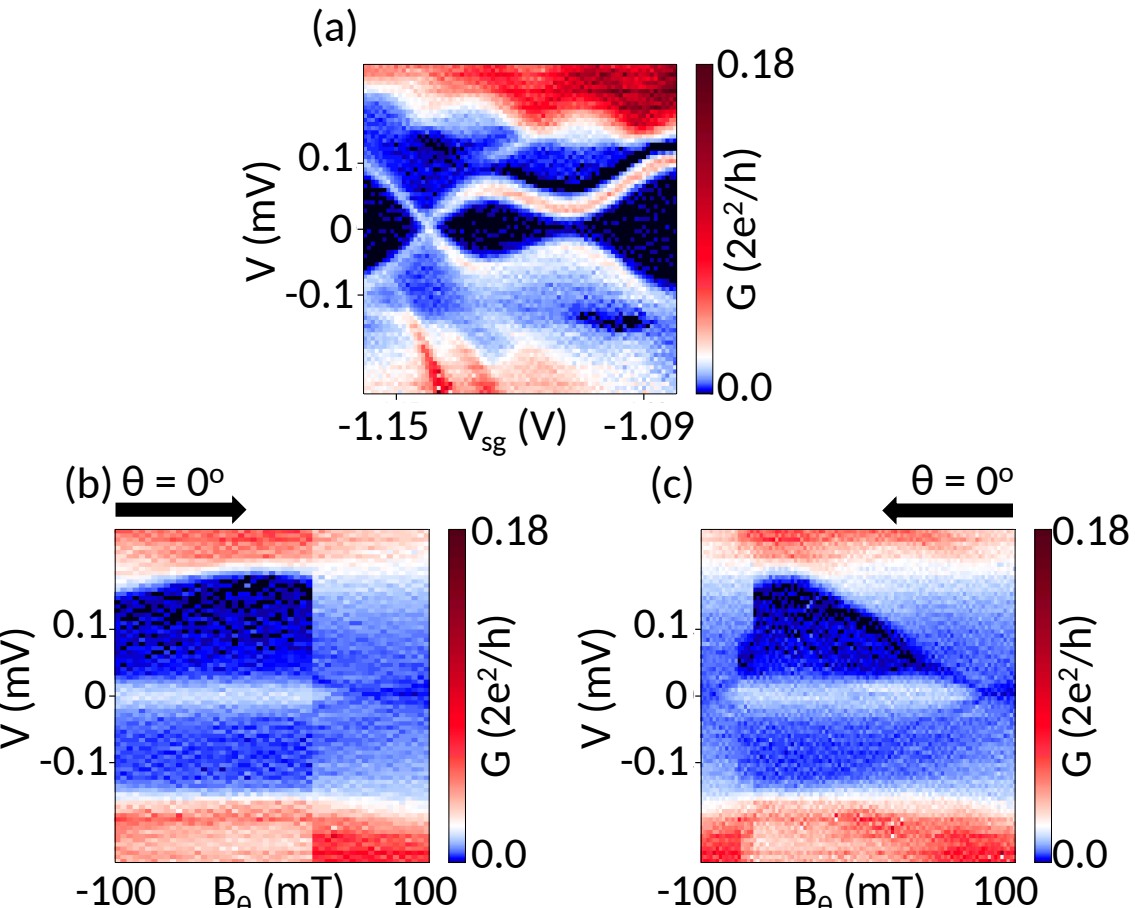

Figure 13: ZBP and hysteresis under magnetic field scans. $V_g = 10.5$ V (a) 2D plots of voltage bias ($V$) with respect of side-gate voltage ($V_{sg}$) in device 3. Conductance is acquired by taking differential of measured current. The CoFe strips are pre-magnetized by applying a -0.1 T magnetic field $B_\theta$ where $\theta = 0^o$. After pre-magnetization, the external magnetic field is swept back to zero for taking the data. (b, c) 2D plots of voltage bias ($V$) with respect of the external magnetic field, $B_\theta$ ($\theta = 0^o$) in device 3. Sweep direction of the magnetic field is given by arrows.

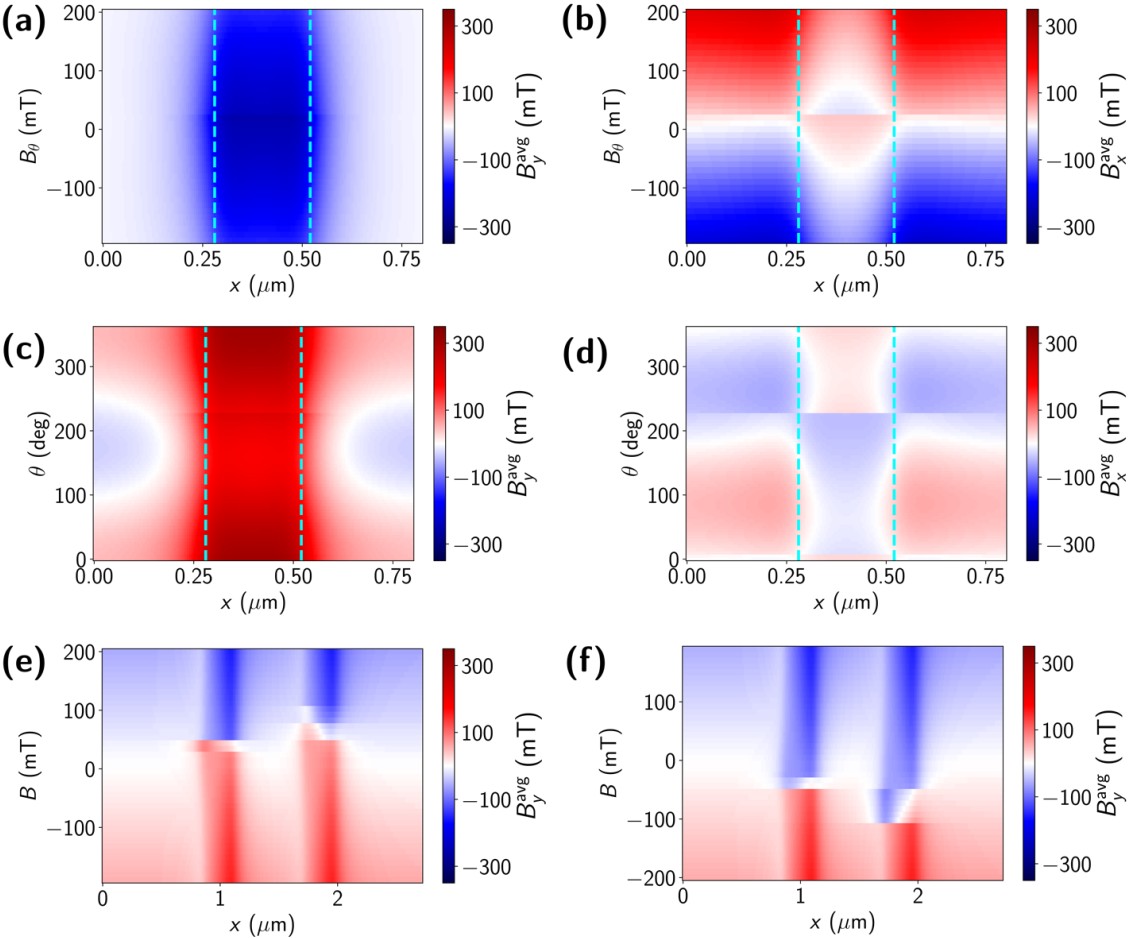

Figure 14: Colormap of (a) $\mathbf{B_y^{avg}}$ (b) $\mathbf{B_x^{avg}}$ as a function of spatial dimension x and $\mathbf{B}_\theta$, with $\theta = 90°$. Colormap of (c) $\mathbf{B_y^{avg}}$ (d) $\mathbf{B_x^{avg}}$ as a function of spatial dimension x and $\theta$ with the magnitude of the applied field $\mathbf{B = 40\ mT}$. Colormap of $\mathbf{B_y^{avg}}$ as a function of $\mathbf{B}_\theta$ and $\mathbf{x}$ for (e) forward and (f) backward sweep with $\theta = 107°$. Dashed cyan lines indicate the location of the CoFe bar magnets.

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
