# Peer review of "Zero-bias conductance peaks at zero applied magnetic field due to stray fields from integrated micromagnets in hybrid nanowire quantum dots"

_SciPost Physics, doi:SciPost Phys. 19, 030 (2025)_

## Round 1 · Referee Report · Helene Bouchiat (Referee 1) · 2025-4-4

Report

This paper presents interesting data on the magneto-transport of in hybrid junctions based on semiconducting nanowires with important Rashba spin-orbit interactions, coupled to superconducting electrodes. These devices have been considered as possible good candidates for the investigation of the physics of Majorana modes when a magnetic field is applied along the nanowires axis or when they are covered by an insulating magnetic layer in both cases opening a spin-orbit gap. The present experiments rely on the implementation of nanomagnets in close vicinity of the nanowires which create stray magnetic fields on the hybrid devices.
The motivation of this work is to reveal the effect of the stray field spatial variations of these nanomagnets on the transport properties of the nanowires reproducing previous controversial data considered as signatures of the existence of Majorana modes and topological superconductivity. Among these signatures is the existence of zero bias differential conductance peaks which are expected to be exactly quantised to the value of e2/h, half of the conductance quantum and was considered as a “smoking gun” proving the existence of Majorana modes in InAs an InSb nanowires. The paper is organised as follows:
-1The authors first present data on a device connected at each end to aluminium electrodes showing the existence of magnetic hysteresis in the critical current in relation of the stray field induced by the nanomagnets.
-2 A second sample connected to one normal and one superconducting electrode, exhibits a zero bias conductance peak between e2/h and 2e2/h at zero magnetic field which is split depending on the gate voltage. Similar features are observed in finite field depending on its orientation and the the magnetic history of the nanomagnet.
-3 Other examples are shown in the supplemental material section. The authors show that, by tuning the magnetic field environment and gate voltage, it is possible to observe a zero bias conductance peak whose amplitude is very close the theoretically predicted value for Majorana modes, even if the investigated devices are in a range of chemical potential and magnetic field parameters where this physics of topological superconductivity is not relevant. These results are important and show the extreme difficulties to obtain conclusive experimental results in this field.
-4 The authors finally emphasize the important fact that the magnetic texture around the wires can be designed in principle at will, using micromagnetic simulations. The obtained results therefore demonstrate the possibility to use nanomagnets to control precisely the magnetic field environment of hybrid NS nano-devices and offer a promising tool to investigate their potentialities concerning the observation of Majorana modes and topological superconductivity.
In that sense this paper “opens a new pathway in an existing or a new research direction, with clear potential for multi-pronged follow-up work”
-5 However at this stage even if the points risen by the authors are very important and justify publication of this work, it would have been nice to have a more precise discussion of the reproducibility of the results obtained within one device and the variability of the results obtained in the different investigated devices. I strongly encourage the authors to provide such a synthesis of their work before publication in SciPost.

Requested changes

Even if the points risen by the authors are very important and justify publication of this work, it would have been nice to have a more precise discussion of the reproducibility of the results obtained within one device and the variability of the results obtained in the different investigated devices. I strongly encourage the authors to provide such a synthesis of their work before publication in SciPost.

Recommendation

Ask for minor revision

  • validity: high
  • significance: high
  • originality: high
  • clarity: good
  • formatting: perfect
  • grammar: perfect

Author:  Sergey Frolov  on 2025-05-23  [id 5516]

(in reply to Report 1 by Helene Bouchiat on 2025-04-04)
Category:
answer to question

We thank Prof. Bouchiat. In response to your comment, we provide an expanded Duration and Volume of Study section:

Duration and Volume of Study
This article is written based on more than 5500 datasets from 5 separate dilution refrigerator cooldowns. For each cooldown, we measure several devices. We measured 19 SN devices, and 20 SNS devices (7 of 20 devices have CoFe strips). Among them, 4 SN devices exhibit clear ABSs; 2 SNS devices show ABSs; and 4 SNS devices have measurable supercurrent. The yield is limited by fabrication issues, device quality and occasional static discharge damage to devices. Within each device, the behaviors in the main text require fine-tuning. The more general behaviors are presented in the supplementary materials.

---

## Round 1 · Referee Report · Pavel Ostrovsky (Referee 2) · 2025-4-17

Report

The paper reports an experimental study of the tunneling spectroscopy on SN and SNS junctions. The junctions are made with InSb nanowires known for their strong spin-orbit coupling and incorporate micromagnets that produce strong local magnetic fields. The main objective of the experiment is to study zero-bias features in the tunneling conductance and evaluate their possible alternative explanations in terms of either topological Majorana or usual Andreev bound states. Experimental studies of the anticipated Majorana bound states in the SN junctions with strong spin-orbit coupling and broken time-reversal symmetry has at least a decade-long history full of hot debates and sometimes controversial claims. The present paper continues this discussion and demonstrates certain similarities in experimental manifestations of the Majorana and Andreev bound states that, if not analyzed properly, may lead to misinterpretation of the results.

Majorana bound states have a topological nature and hence are to a certain extent robust with respect to variations of system parameters. In particular, they always occur at exactly zero energy near the SN interface and under ideal conditions should be visible as zero-bias peaks of the height of $e^2/h$ in the tunneling conductance. Under realistic conditions tunneling conductance peak may be of a smaller magnitude. Experimental setup reported in the paper was intentionally designed such that the Majorana states do not appear. It is demonstrated that, with a proper tuning of parameters, a zero-bias peak in the tunneling conductance is still observed with the magnitude either smaller or larger than $e^2/h$. In certain cases the zero-bias conductance exhibits a relatively long plateau near the desired value of $e^2/h$ as a function of the side gate voltage.

The topic of the paper is very interesting and concerns a rapidly developing field of modern research. The accomplished thorough experimental study is definitely worth publishing. There are however several details in the presentation of the experiment and its results that should be clarified.

Requested changes

  1. In the introductory part of the paper (page 2) there is the following statement about Majorana bound states: "The quantization is rapidly lost with departure from ideal conditions." This statement is at least misleading. In fact, the major scientific interest in the Majorana bound states is due to their topological origin, which provides a certain degree of robustness with respect to external conditions. It is exactly this robustness, that is required to realize quantum computations with the Majorana states. The authors should rephrase their statement in a way that would not lead to misunderstanding. It is also necessary to explain in more detail the distinction between topological Majorana and usual Andreev bound states and, if possible, formulate some criteria to distinguish them. More on this in the next item.

  2. The authors claim that Majorana bound states exhibit a zero bias peak of at most $e^2/h$ while usual Andreev states can have higher tunneling conductance. On page 2, it is written that the zero-bias peak due to usual Andreev bound states can get larger than $2e^2/h$. Factor of two here is probably a misprint. A single bound state cannot provide such a high conductance. On the other hand, if two Majorana bound states located e.g. at opposite ends of an SNS junction have a small but for some reason non-negligible overlap (normally the overlap should be exponentially small with the distance between them), they will hybridize into a single normal Andreev bound state with a small but nonzero energy. What would be than the magnitude of the conductance peak? Will it abruptly grow from $e^2/h$ to $2e^2/h$ once the small overlap is taken into account (given that all other conditions are ideal)? What will happen if the width of this peak (e.g. due to temperature) is larger than the energy of the Andreev state itself and the peak visually occurs at zero energy? This point needs some more explanation.

  3. According to the statement on page 4, the device 1 is fabricated with two junctions but the results are analyzed as if there was only one junction. Why does it work like this?

Recommendation

Ask for minor revision

  • validity: -
  • significance: -
  • originality: -
  • clarity: -
  • formatting: -
  • grammar: -

Author:  Sergey Frolov  on 2025-05-23  [id 5517]

(in reply to Report 2 by Pavel Ostrovsky on 2025-04-17)
Category:
answer to question

We thank Dr. Ostrovsky. We provide discussion and address requested changes.

  1. In the introductory part of the paper (page 2) there is the following statement about Majorana bound states: "The quantization is rapidly lost with departure from ideal conditions." This statement is at least misleading. In fact, the major scientific interest in the Majorana bound states is due to their topological origin, which provides a certain degree of robustness with respect to external conditions. It is exactly this robustness, that is required to realize quantum computations with the Majorana states. The authors should rephrase their statement in a way that would not lead to misunderstanding.

We agree that the notion of conductance quantization is misleading, because really it is only expected in an infinitely long nanowire with zero disorder and at zero temperature. In fact, the paper that popularized the idea to search for quantized conductance has a dramatic calculation in its Figure 5 that shows how quickly any quantization is erased when realistic terms are added to the Hamiltonian.

We modify the sentence the following way: “The quantization is rapidly lost with departure from ideal conditions (see Figure 5 in ref. [22]).”

It is also necessary to explain in more detail the distinction between topological Majorana and usual Andreev bound states and, if possible, formulate some criteria to distinguish them. More on this in the next item.

We added a more background to discuss ABS on page 2. It is an open problem as to what are the specific criteria for identifying MZM and distinguishing them from ABS. This question will have more practical value when real candidate MZM are identified and the data cannot be ruled out as trivial ABS by simply considering a slightly larger set of data than what is shown in a paper.

  1. The authors claim that Majorana bound states exhibit a zero bias peak of at most e2/he2/h while usual Andreev states can have higher tunneling conductance. On page 2, it is written that the zero-bias peak due to usual Andreev bound states can get larger than 2e2/h2e2/h. Factor of two here is probably a misprint. A single bound state cannot provide such a high conductance. On the other hand, if two Majorana bound states located e.g. at opposite ends of an SNS junction have a small but for some reason non-negligible overlap (normally the overlap should be exponentially small with the distance between them), they will hybridize into a single normal Andreev bound state with a small but nonzero energy. What would be than the magnitude of the conductance peak? Will it abruptly grow from e2/he2/h to 2e2/h2e2/h once the small overlap is taken into account (given that all other conditions are ideal)? What will happen if the width of this peak (e.g. due to temperature) is larger than the energy of the Andreev state itself and the peak visually occurs at zero energy? This point needs some more explanation.

Zero bias peaks due to MZM in nanowires measured by tunneling can reach conductance values of 2e2/h or they can be shorter. But they cannot be taller than 2e2/h when a basic Majorana model is used (See Figure 1 in Reference 22). The factor 2 is due to resonant Andreev reflection and particle-hole symmetry. Zero-bias peaks due to ABS can be both taller and shorter than 2e2/h. One way this can happen is, indeed, when multiple ABS coalesce near zero bias. In Figure 3 we subtracted a particular series resistance to plot the plateau-like zero-bias peak near e^2/h, which is an allowed value for Majorana, however it is below the quantized Majorana conductance value. This is done to illustrate that plateaus can appear at different values of conductance and by themselves do not indicate the presence of MZM. In this paper we do study SNS junctions but we only study supercurrents in them. We study zero bias peaks in what nominally are NS junctions, so we do not expect coupled MZM across the junction. When two MZM are hybridized across the junction there will be no tunneling barrier to detect the zero-bias peak if a two-terminal geometry is used. If another lead is added for tunneling near the junction, then in the simplest Majorana model the detected peak would transition from a quantized value of 2e^2/h to a new value determined by MZM coupling, tunnel coupling and temperature.

  1. According to the statement on page 4, the device 1 is fabricated with two junctions but the results are analyzed as if there was only one junction. Why does it work like this?

We do not have enough control knobs in this particular experiment to answer the question ‘why’. For instance, we only have one single gate electrode so we cannot selective address the two junctions.

In the context of this specific study this question is not central because the point here is to show examples of data that challenge exotic narratives, and we do have enough information to rule out such narratives, e.g. that the hysteretic critical current is not because of the exchange interaction induced in the semiconductor by the ferromagnet.

By way of speculation, one possibility is that the left junction is too close to the gold lead, and it is poisoned by the normal metal so that the critical current of that junction is both suppressed and does not manifest as a sharp switch. Since these are two-terminal measurements the normal state resistance of that junction is simply folded into the series resistance present in the circuit and subtracted out.

---

## Round 2 · Referee Report · Pavel Ostrovsky (Referee 2) · 2025-5-27

Report
Recommendation
Publish (easily meets expectations and criteria for this Journal; among top 50%)

---

## Round 2 · Referee Report · Helene Bouchiat (Referee 1) · 2025-6-9

Report
I recommend this paper for publication in its present form.
Recommendation
Publish (easily meets expectations and criteria for this Journal; among top 50%)

---

## Round 2 · Author Response

List of changes
-
We provide an expanded Duration and Volume of Study section
-
We modify the sentence the following way: “The quantization is rapidly lost with departure from ideal conditions (see Figure 5 in ref. [22]).”
-
We added a more background to discuss ABS on page 2.

---

## Round 2 · List of Changes

-
We provide an expanded Duration and Volume of Study section
-
We modify the sentence the following way: “The quantization is rapidly lost with departure from ideal conditions (see Figure 5 in ref. [22]).”
-
We added a more background to discuss ABS on page 2.

---

## Editorial Decision

published